# New Developments in Medical Applications of Hybrid Hydrogels Containing Natural Polymers

**DOI:** 10.3390/molecules25071539

**Published:** 2020-03-27

**Authors:** Cornelia Vasile, Daniela Pamfil, Elena Stoleru, Mihaela Baican

**Affiliations:** 1Physical Chemistry of Polymers Department, “P. Poni” Institute of Macromolecular Chemistry, 41A Gr. Ghica Voda Alley, RO, Iaşi 700484, Romania; pamfil.daniela@icmpp.ro (D.P.); elena.paslaru@icmpp.ro (E.S.); 2Pharmaceutical Physics Department, “Grigore T. Popa” Medicine and Pharmacy University, 16, University Str., Iaşi 700115, Romania

**Keywords:** organic hybrid polymeric hydrogels, natural polymers, medical applications, homopolysaccharides, heteropolysaccharides, polypeptides, proteins

## Abstract

New trends in biomedical applications of the hybrid polymeric hydrogels, obtained by combining natural polymers with synthetic ones, have been reviewed. Homopolysaccharides, heteropolysaccharides, as well as polypeptides, proteins and nucleic acids, are presented from the point of view of their ability to form hydrogels with synthetic polymers, the preparation procedures for polymeric organic hybrid hydrogels, general physico-chemical properties and main biomedical applications (i.e., tissue engineering, wound dressing, drug delivery, etc.).

## 1. Introduction

Hydrogels can be classified by taking into consideration many factors, such as source; preparation methods; network structure (as permanent (chemically crosslinked or irreversible), and non-permanent (physically crosslinked or reversible, hydrogen-bonded hydrogels); dimensions (macrogels, microgels, nanogels); sensitivity to stimuli (such as physical, chemical, and biochemical stimuli); charge of polymer network (nonionic, ionic, zwitterion, and amphoteric); physical aspect (micro-/nanoparticle, film, matrix, gel, etc.); configuration (amorphous and semicrystalline); composition (homopolymeric, multipolymeric or heteropolymeric, copolymeric, and interpenetrating polymer networks, hybrids, composites); degradability (biodegradable, bioabsorbable, bioerodible, and degradable in a controlled manner) (Scheme 1) [1,2].

Generally, hydrogels contain polar/charged functional groups which offer them hydrophilicity, water absorption capacity and, respectively, swelling in a certain medium, enhancement of their susceptibility to stimuli, etc. [3,4]. They can also differentiate in respect with their equilibrium swelling grade (SWD) as those low SWD hydrogels (20–50%), medium SWD hydrogels (50–90%), high SWD hydrogels (90–99.5%), and superabsorbent hydrogels (>99.5%) [5,6]. The hydrogels with high SWD show good permeability and biocompatibility [7] being preferred for use in the medical field.

Hybrid hydrogels definition is still debatable. They are defined either as a complex composed of hundreds of chemically or physically cross-linking nanogels [8], or it refers to systems combined with different polymers and/or with nanoparticles, such as plasmonic, magnetic, and carbonaceous nanoparticles, among others, or they are constituted by chemically, functionally, and morphologically distinct building blocks from at least two distinct classes of molecules, which can include biologically active polymers as polysaccharides and/or proteins, peptides, or nano/microstructures, interconnected via physical or chemical means [9]. Depending on the size and the nature of the building blocks, the hybridization can occur at molecular level or at microscopic scale [10,11].

For the purpose of this review, we refer only to the organic polymeric hybrid hydrogels containing natural polymers (Figure 1), defined according to the last definition and their medical applications (in medicine/nanomedicine).

Each medical application involves the unique choice of a combination of the component materials, with the goal to match both desired structural and functional properties which must effectively produce an advanced polymeric system, with a new profile [12]. One of the most relevant examples is the combination protein/other polymers. Such combinations can be resulted by polymerization or conjugation (click chemistry) with synthetic polymers resulting compatible hybrid hydrogels both in vitro and in vivo as it was demonstrated by cell differentiation, proliferation, migration studies and drug delivery, tissue engineering, wound healing applications [13,14], respectively or sequestration of growth factors from the surrounding medium [15]. Commonly, the hybrid hydrogels are heterogeneous and this property is important to assure cell adhesion, organization, and cell–cell interactions required for medical applications [16,17,18,19].

### 1.1. Polymers Used in Hybrid Hydrogels

There are four main types of natural biodegradable polymers used in hybrid hydrogels described in this review—Table 1, including [20]: (1) homopolysaccharides, as: cellulose and derivatives, pullulan, dextran, starch, etc.; (2) heteropolysaccharides from which can be mentioned: chitosan/chitin and their derivatives [21], dextran, agarose, alginic acid and alginates, hyaluronic acid (HA), chondroitin and derivative sulphates, heparin, pectin, etc. (3) polypeptides/proteins, such as gelatin, collagen, albumin, fibrin and fibrinogen, soy and whey proteins, silk, Matrigel™, etc., and genetically engineered proteins [22,23,24] (calmodulin (a calcium-binding protein), elastin-like polypeptides, leucine zipper) [25]; (4) deoxyribonucleic acid (DNA) and ribonucleic acid (RNA) [26]. The protein/polysaccharide hybrid polymers like fibrin/cellulose, collagen/HA, gelatin/alginate and many others etc. were studied [27] and other many combination make now topics of undergoing researches. Lignin was also used [28,29]. Most of them are components of the extracellular matrix (ECM) in vivo. Their composition (bovine fibrinogen, rat tail collagen, etc.) may vary with source and processing method, being difficult to control their microstructures, properties and reproducibility between experiments.

Synthetic polymers commonly used in the hybrid hydrogels preparation can be classified into three main types: non-biodegradable [30,31], biodegradable [32], and bioactive polymers [33]. Most common synthetic polymers are: poly (lactic acid) (PLA), poly (ε-caprolactone) (PCL), poly(glycolic acid) (PGA) and copolymers [34], poly (ethylene glycol) (PEG) and poly(vinyl alcohol) (PVA) [35,36,37,38] to produce biodegradable hydrogels. Hydrogels may include vinyl monomers like 2-hydroxyethyl methacrylate (HEMA), N-isopropyl acrylamide (NIPAAm), 2-hydroxypropyl methacrylate (HPMA), acrylamide (AAm), acrylic acid (AAc) or macromers [37,38,39], methoxyl poly(ethylene glycol) (PEG), monoacrylates (mPEGMA or PEGMA), and diacrylates (PEGDA), ethylene glycol diacrylate (EGDA), Pluronic^®^ polymers, etc. [39].

By combining the properties of synthetic and natural polymers to form hybrid hydrogels, a direct approach is created for bioactive hydrogel scaffolds for tissue engineering.

Comparatively with natural polymers, the synthetic polymers are easily synthesized even at large scale by polymerization, cross-linking, and functionalization (modification by block structures, by blending, copolymerization), their molecular structure, molecular weight, physical and chemical properties (mechanical strength, biodegradability [40,41]) are more reproducible, this aspect being critical for the medical applications mainly scaffolding. Unfortunately, applications of synthetic hydrogels as biomaterials are limited by their absence of bioactivity. The protein-polymer hybrid networks with complex abilities, including bioactivity, stimuli-responsiveness, catalytic activity, or ability to regulate cell behaviors have been/are created to overcome this limitation, maintaining good mechanical properties of materials [42,43,44,45,46].

#### 1.1.1. Microgel

The term microgel describes a variety of particles that differ substantially in structure, physico-chemical properties, preparation and application and is interchangeably with terms such as nanogel, microsphere and macrogel depending on the numerous particle types falling within the broad sphere of nano-/microparticle shapes and sizes [47,48,49,50].

#### 1.1.2. Hybrid Nanogels

Hybrid nanogels later developed are highly crosslinked nano-sized hydrogel systems [47,48] with diameter less than 100 nm [49,50] having a non-fluid colloidal/polymer network that combine the properties of both hydrogels and nanomaterials. The nanoscale provides a large surface area for bioconjugation, long time of circulation in blood, and the possibility of being actively or passively targeted to the desired site of action (e.g., tumor sites) [10]. Hybrid smart hydrogels/nanogels show the ability to respond to biomedically relevant changes like pH, temperature, ionic force/concentration, redox environment, light, glucose, magnetic field, electrical field, chemicals or specific biomarkers etc., by changing their volume, refractive index, and hydrophilicity/hydrophobicity etc. Micro- and nano-sized hydrogels are faster in responding to changes in their environment than their macroscopic or bulk counterparts and can be used more efficiently in medical and sensor applications [51].

#### 1.1.3. Multifunctional Hybrid Nanogels

Multifunctional hybrid nanogels found applications in medical field/nanomedicine for continuous monitoring by optical sensing to mentioned stimuli in complex samples such as blood and bioreactor fluids as well as for intracellular imaging, contributing to the explanation of intricate biological processes, the development of novel diagnoses and therapy toward clinical applications. [52].

#### 1.1.4. Hybrid Polymer Nanogel/Hydrogels

Hybrid polymer nanogel/hydrogels include interpenetrated networks (IPNs) and core-shell particles. The core-shell strategy is especially useful for targeting therapy, while the interpenetration allows the development of multiresponssive nanogels and the control of the drug release profile [53].

#### 1.1.5. Physical Hydrogels

Physical hydrogels result by ionic and physical interactions, such as hydrogen bonds, coordination bonds, electrostatic and hydrophobic interactions in certain conditions and physico-chemical interactions (stereo-complexation, charge condensation, or supramolecular chemistry) [54]. By changing the temperature, pH, ionic strength or solvent composition, they form a homogeneous solution and re-gel when they return to their initial conditions, being reversible gels, generally unstable and mechanically weak [55]. The physical cross-links are also formed by crystallization, [56] between amphiphilic block and graft copolymers [57], and protein interactions [58]. Physically crosslinked hydrogels show stimuli-responsiveness and self-healing properties, but their mechanical strength is low and they often exhibit plastic flow [59].

#### 1.1.6. Chemically or Covalently Crosslinked Hydrogels

Chemically or covalently crosslinked hydrogels with a permanently fixed shape at rest, exhibit a low fracture toughness and extensibility. Therefore, it is preferred to create both physically and covalently crosslinking hydrogels [60,61], resulting doubly-crosslinked hybrid gels that combine all mentioned properties [62]. Many double network (DN) hydrogels prepared by double chemically crosslinking or by hybrid physical/chemical crosslinking imply crosslinking agents, but they present toxicity which is an important disadvantage. Designing a new generation of DN gels comprising two non-covalent associated networks is a promising technique.

Kondo and coworkers [63] prepared a dually-crosslinked polymer gel with a very homogeneous network architecture, using a tetra-arm star-shaped poly(ethylene glycol) (PEG), PEG and poly(dimethylsiloxane) (PDMS) building blocks linked by orthogonal cross-coupling, The obtained network from hydrophilic and hydrophobic components regularly and uniformly distributed is non-covalent hydrophobic association whose strength is tuned by the molar ratio of the hydrophilic PEG and the hydrophobic PDMS segments [64].

#### 1.1.7. Self-Assembling Hybrid Hydrogels

Self-assembling hybrid hydrogels containing peptides provide the desired biological functionality and biodegradability, are able to mimic biological structures and materials having direct biomedical applications, namely as carriers for drug and cell delivery (e.g., incorporation of bioactive sequences from natural proteins). To control mechanical, biocompatibility and degradation properties, the peptides are combined with polymeric networks [65,66] by chemical modification, covalently linking or non-covalent interactions between peptides and polymers [67].

Hybrid hydrogels self-assembled from graft copolymers via formation of coiled coil antiparallel heterodimers was also demonstrated [68], based on HPMA copolymers backbone and a pair of oppositely charged peptide grafts. The formation of these hybrid hydrogels was reversible [68]. A DNA/poly(lactic-*co*-glycolic acid) (PLGA) hybrid hydrogel (HDNA) was prepared for water-insoluble ophthalmic therapeutic delivery of dexamethasone and it may be applied in treatment of various eye diseases [69].

#### 1.1.8. Interpenetrated and Semi-Interpenetrated Polymer Networks

To enhance the mechanical strength, the swelling/deswelling response, and to add new sensitivities to a nanogel, multicomponent networks as full IPNs and semi-IPNs (sIPNs) were prepared by simultaneous synthesis and sequential synthesis involving two or more polymers [70,71]. The reaction can take place in the presence of a crosslinking agent, in order to form a complete IPN or in the absence of the crosslinking initiator, to form a sIPN.

#### 1.1.9. Core-Shell Polymer Networks

The most common techniques of synthesis of core-shell nanogels are the seed precipitation polymerization, crosslinking of amphiphilic micelles preformed by self-assembly or the reversible addition–fragmentation chain-transfer polymerization (RAFT) [72,73,74,75,76,77].

Several examples of hybrid polymeric hydrogel include:
(1)PEG-modified natural polymers [11,78,79], like fibrinogen, heparin (Hep), dextran, HA, and albumin;(2)PNIPAAm-modified natural polymers, like collagen, chitosan, and alginate [80,81,82,83].


#### 1.1.10. Supramolecular Hydrogel

Supramolecular hydrogel are builded by blocks of peptides and polymers by the coupling/conjugation of specific peptide sequences (cell adhesive and/or enzymatically cleavable) to polymer chains. In such a way is obtained controlled cell responses (adhesion, migration, differentiation) because the components can self-assembly into hybrid hydrogels either, as peptide-polymer conjugates or combining individual components. These will determine the properties of the hydrogels (as stiffness, mesh structure, responsiveness, and biocompatibility) [84], cooperative folding/unfolding transitions control over the structure formation at the nanometer level. The new produced materials may possess unprecedented levels of structural organization and novel properties [85]. By optimizing the amino acid sequence, responsive hybrid hydrogels tailor-made for a specific application may be designed. Hybrid peptide/polymer molecular hydrogel design and synthesis showed significant research progress to mimic the natural proteins molecular architectures, dynamic responsiveness, and cellular functions, combined with tunability and processability provided by the synthetic polymer constituents.

## 2. Preparation Procedures for Polymeric Hybrid Hydrogels

### 2.1. Routes to Obtain Hybrid Hydrogels

Crosslinking techniques can be: (i) physical crosslinking (achieved by using repeated freezing/thawing cycles and led to cryogels) by ionic interaction, complex coacervation or H-bonding; (ii) chemical crosslinking or grafting by polymerization, co-polymerization, chemical conversion (using crosslinking agents such as borates, glyoxal, glutaraldehyde, etc.), and (iii) irradiation crosslinking or grafting (electron beam or gamma radiation, depending on irradiation dose). The properties of hydrogels can be controlled by different parameters, such as structures, by cross-linking type, end density, and synthesis of polymers, while in the case of physical hydrogels, by environment conditions (as pH, temperature, ionic strength etc.).

Chemically cross-linked gels are obtained by radical polymerization/crosslinking, emulsion, reverse microemulsion, inverse miniemulsion, heating, irradiation (ultraviolet, high-energy radiation, especially gamma and electron beams), photolithographic chemical reactions via crosslinker as di-sulfide crosslinking, ionic, click chemistry (such as azide-alkyne cyclo-addition reactions, thiol-ene couplings, Diels-Alder reactions and tetrazine-norbornene chemistry), Schiff base crosslinking with a huge ensemble of reactions, such as Michael type reaction, Michaelis-Arbuzov reaction, and nucleophile addition [86], and enzymatic cross-linking [87]. Both chemical and physical cross-linking approaches are employed for hydrogels preparation [2].

A breakthrough toward the synthesis of complex structures with a high degree of functionality and compositional variety is the utilization as synthesis ways the controlled/living radical polymerization technique such as the catalytic atom (group) transfer radical polymerization (ATRP), degenerative chain transfer polymerization represented by iodine-mediated polymerization (RITP), and reversible addition-fragmentation chain transfer polymerization (RAFT) [88]. A new strategy of hybrid hydrogels synthesis entails the non-covalent attachment of genetically engineered coiled-coil protein motifs to hydrophilic synthetic HPMA copolymer backbone. The physical crosslinking was established by self-assembly of the coiled-coil domains [89].

#### 2.1.1. Chemical Modifications

Chemical modifications involve a plenty of ligands which can be used for targeted drug delivery, stimulus responsive drug release or preparation of complex materials. The cross-linking of the hybrid network and conjugating proteins to the gel backbone as a platform for immobilizing functional proteins was reported by Lim et al. [90].

#### 2.1.2. Functionalization

Hybrid hydrogels/nanogels can also be surface functionalized with specific ligands to achieve targeted therapy and reduce toxicity [91]. Functionalization is also important in order to create different types of macro/micro/nanogels morphologies, as hairy microgels, core-and-shell, hallow, multilayer microgels, [92] etc.

#### 2.1.3. Stealth Functionalization

Hybrid nanosystems/nanogels for drug delivery and biomedical purposes need a non-secondary requirement, as their biocompatibility necessary both to reduce the inflammatory or the immune response of the organism, and to improve blood circulation lifetime, biodistribution, and bioavailability of the carried drugs and also to overcome the self-defense mechanisms present in the bloodstream of the host organism. To achieve this requirement the hybrid nanogels must be specifically designed. A very wide variety of architectures result by their decoration, modification, and functionalization, [93], or they can be modified by conjugation with both organic [94] and inorganic [95] types of nanoparticles and nanostructures. The morphologies of hybrid nanogels vary both with the particle type and the assembly technique, each component being either core or shell, of different size and architecture [96]. These variable morphologies may be obtained by chemical reactions or through physical crosslinking based on hydrogen bonds, ionic interactions, and other intermolecular bonds. Therefore, a proper surface decoration and its biocompatibility, is a parameter capable of strongly influencing the biodistribution together with the dimensions, the surface charge and the ligands interaction. Many stealth functionalizations exploit hydrophilic polymeric chains, as polyethylene glycols or chitosan.

#### 2.1.4. PEGylation

PEGylation is a solution to increase the bioavailability of the decorated nanostructures and to extend the circulating lifetime [97]. After this modification a protein corona is formed around the antifouling PEG functionalization [98]. It will create a hindered zone around the nanoparticles and reduces the wrapping by plasma proteins and the subsequent uptake by macrophages PEGylation depends on many factors such as hydrophilicity of the PEG chains, molecular weight (MW) which vary from 2000 to 13,000 Da.

### 2.2. Processing Methods

Processing methods include [1]: solution casting/drying, theta gelation, freezing or freezing/pressurizing, freeze drying, emulsion freeze drying, inverse microemulsion polymerization technique, solution blowing, electrospinning, coagulation treatment, CO_2_-in-water emulsion, sol-gel method/thermal annealing, CO_2_ bubbles template freeze drying, high hydrostatic pressure [HHP] method, supercritical gel-drying. Other new synthesis methods include the implementation of click chemistry reactions [99], photo-patterning, and rapid prototyping, 3D printing for the facile production of hybrid hydrogels, self-assembly [100,101], the use of biological molecules and motifs to promote a desired cellular outcome, and the tailoring of kinetics and transport behavior to obtain desired biomedical outcomes [102]. 3D bioprinting of hydrogels is performed in accordance with the native tissue architecture therefore it is expected to result in a new generation of engineered tissues. Bakarich et al. [103] fabricated by a new 3D-printing approach an interesting material with good mechanical performance based on κ-carrageenan and poly(oxyalkylene amine) (Jeffamine) based ionic-covalent entanglement hydrogels. The carrageenan induced a fast gelation, a structural integrity to the hydrogel system and thermoresponsiveness, while the epoxy-amine reaction to form covalent bonding takes place an ambient temperature for covalent bond formation.

Hydrogels and their products can be obtained in a wide range of shapes as temporary or permanent shape, shape memory, smart shape memory, quadruple-shape, sponges, soft or rigid, stretchable, films, sheets, bilayer, micro/nanoparticles with defined shapes, ultrathin microcapsules, matrix, scaffolds, hollow cube, hemisphere, pyramid, cylindrical, twisted bundle, patches for wound dressing, artificial ear, nose, and many others.

## 3. Properties

The specific physico-chemical key properties of the hybrid hydrogels are: remarkable thermodynamic stability, elevated capacity of solubilization, mildness, density, swelling/deswelling, high-water content and permeability, low surface tension and relative low viscosity, stiffness, mesh structure and size, responsiveness, biocompatibility and biodegradability (so avoiding its accumulation in the organs), non-immunologic response and capability of undergoing vigorous sterilization techniques [48], as well as their tunable viscoelasticity and structural similarity to the ECM. Their properties can be fine-tuned through selection of the hydrogel components (chemical composition), hydrophobicity/hydrophilicity ratio, and cross-linking strategy, crosslinking density etc. Hydrogels are commonly considered as highly biocompatible, owing to the high-water content and also to the physico-chemical similarity with the native ECM. Chemically cross-linked synthetic polymeric hydrogels have higher mechanical properties compared to self-assembling (physically crosslinked) systems, thanks to the high molecular weight of polymer materials, but they lack biological functionality, while self-assembling hydrogels, formed through physical cross-links, allow minimally invasive implantation in the body.

### 3.1. Swelling

The swelling of hydrogels is a process occurring in three steps, namely: (a) diffusion of water molecules into hydrogel network, (b) hydration of polymeric chains and their relaxation and (c) expansion of crosslinked polymeric network. The primary and secondary bound water is uptaken by the network by its interaction with the polar and hydrophobic sites, respectively and then the network is imbibed with additional water which is named free water. Finally at an infinite dilution to a maximum, level equilibrium water content is reached. The determination of swelling behavior is the main assay to establish the hydrogel quality, as it is also a means to evaluate other properties as: crosslinking degree, mechanical properties, degradation rate, etc. Swelling properties of the stimuli responsive hydrogels are significantly changed by the modification in parameters of the surrounding environment (i.e., temperature, pressure, pH, solvent composition, ionic strength, electrical potential, etc.). The polymeric hybrid hydrogels exhibit biodegradability and biocompatibility, high permeability, to oxygen, nutrients, and to water-soluble metabolites, being promising carriers and for cells encapsulation. They resemble with natural soft tissues [41,104] being very useful in regenerative medicine, for tissue scaffold or therapeutic transfer systems, promoting cell attachment and proliferation [2].

### 3.2. Mechanical Properties

The mechanical properties can be varied and tuned by changing the crosslinking degree, or lowered by heating. To seed osteoblast cells, it is necessary a more stiff material than in the case of adipocyte culture, as for this is also requirement for the development of a heterogeneous prosthetic device, as substitute for the intervertebral disc. The elastic nature of hydrated gels has been found to minimize irritation to the surrounding tissues after implantation.

### 3.3. Responsiveness

Generally, hydrogels have weak mechanical properties and a slow or delayed response to external stimuli. Novel hydrogel designs substantially enhanced mechanical properties and by creating the superporous and comb-type grafted hydrogels fast responses to external stimuli were obtained as also was done by development of self-assembling hydrogels from hybrid graft copolymers with property-controlling protein domains, and genetically engineered triblock copolymers containing hydrogels.

The low interfacial tension between the gel surface and body fluid minimizes protein adsorption and cell adhesion, reducing the chances of negative immune reactions [105].

### 3.4. Porosity and Permeation

The average pore size, the pore size distribution, and the pore interconnections included together in the parameter called « tortuosity » are important factors for a hydrogel matrix. They are influenced by the composition and the crosslink density of the hydrogel polymer network. Pores can show different morphologies: they can be closed, open as a blind end or interconnected, again divided in cavities and throats.

Net charge of the polyelectrolyte hydrogel is determined by the initial concentration of the cationic and/or anionic monomer.

Crosslinking influences all the other properties of the hydrogels. By controlling the crosslinking degree, the materials with tunable and optimized properties destined to different applications can be obtained [106].

The micro-/nanogels are valuable materials as drug-delivery carriers because they show high loading capacity, good stability, and reversible volume change in response to environmental stimuli (such as pH, temperature, and glucose level) [93].

## 4. Applications

Hydrogels remain the most appealing candidates for tissue engineering scaffolds. The development of hybrid hydrogels constituted from different polymers is based on numerous resources and they are applied for regenerative medicine, tissue engineering (including: bone regeneration [107,108,109,110], cartilage tissue, vascular tissue, cardiac tissue, cardiovascular tissue, meniscus tissue, human prostate tissue, skin tissue/wound, and other tissues), wound healing, artificial cornea, drug/gene delivery, cancer cells, nucleus pulposus bioelectronic interfaces due to their structural similarity to the natural ECM, inherent biocompatibility, tunable viscoelasticity, tunable physical and mechanical properties, and their ability to form scaffolds for different tissues, high-water content and high permeability for oxygen and essential nutrients [11]. Biomedical applications of hydrogels as the first materials developed for uses inside the patient started from the decade of 70 s [111].

It is considered that the development of the hydrogels for medical applications known three steps [100,112]. The first generation of hydrogels is characterized by various crosslinking procedures involving the chemical modifications of a monomer or polymer with an initiator to develop materials with high swelling and good mechanical properties. The second generation of materials is that capable to respond to specific stimuli (temperature, pH, ionic strength, different external fields or concentration of specific bioactive molecules etc.), known as smart hydrogels. Finally, the research for the third generation of hydrogels was focused on the investigation and development of hybrid, stereo complexed materials (e.g., PEG-PLA interaction) with a wide spectrum of tunable properties and trigger stimuli [113,114]. This last stage aimed to develop the so called “smart hydrogels” with a variety of possible applications. Hybrid hydrogels based on both natural and synthetic polymers offer infinite possibility to cells encapsulation, as matrices for repairing and regenerating a wide variety of tissues and organs [115], are capable of responding to biological signals in vivo or remote triggers and other many possible applications in biomaterials, biomedicine and nanomedicine [116].

Other important applications are [102] (Scheme 2): wound dressing/healing, treatment of severe burns, drug delivery/controlled release, injectable hydrogels, vaccines, cancer treatment, autoimmune disease, neurodegenerative disease, anti-inflammatory, ophthalmology, etc.

Particularized examples of medical applications of hybrid hydrogels are described in the following sections.

## 5. Homopolysaccharides-Based Hybrid Hydrogels

### 5.1. Ability of Homopolysaccharides to Form Hybrid Hydrogels

Homopolysaccharides (HP) are subdivided into straight chain and branched chain ones, into plant polysaccharides, animal polysaccharides, microbial/bacterial polysaccharides, and seaweed polysaccharides.

Most homopolysaccharides can form hydrogels due to their intrinsic properties and the gel formation is generally driven by physical interactions. Amongst the plant-derived homopolysaccharides, cellulose and its derivatives possess plentiful hydrophilic functional groups (such as hydroxyl, carboxyl, and aldehyde groups) in the backbone that can be used to prepare hydrogels [117]. Starch is the most abundant storage polysaccharide in plants and includes two main structural components, namely amylose and amylopectin. The synthesis of starch hydrogels is determined by important features such as gelatinization and retrogradation, which are in turn affected by amylose and amylopectin ratio [118]. The hydrogels obtained from native starch, pure starch components and their derivatives are hydrophilic and of great significance in the biomedical domain because of their good swelling capacity in water, biocompatibility and biodegradability [119]. Carrageenan (CG) family of polysaccharides are soluble in hot water (>60 °C) and forms thermoreversible gels in a process that is dependent on temperature (when dropped down to 30 °C–40 °C gelation occurs) and the type of ions [120]. Due to the structural resemblance to glycosaminoglycans (GAGs) (that is a component of natural extracellular matrix—ECM) and its fine physical functional properties, CG is extensively used in biomedical applications. Formation of gellan gum (GG) (a linear anionic exopolysaccharide) –based hydrogels takes place in the presence of mono-, di- and trivalent cations and depends on the temperature [121].

### 5.2. Biomedical Applications of Homopolysaccharides-Based Hydrogels

Homopolysaccharides native or modified with the various conjugates have been extensively used to develop organic hybrid hydrogels, for combating last-ditch biomedical challenges. In Table 2 are listed the several examples of components in homopolyssaccharide-based organic hybrid hydrogels, their synthesis pathways and medical applications.

#### 5.2.1. Tissue Engineering

Multicomponent hydrogels based on PHEMA matrix and BC nanofibers were successfully prepared by in situ UV radical polymerization of HEMA monomer impregnated into wet BC nanofibrous structure. Biocompatibility tests demonstrated that BC-PHEMA hydrogels are non-toxic providing a favorable environment for proliferation of marrow stem cells isolated from rabbits (rMSCs)—Figure 2. Therefore, the obtained hydrogels can be seen as promising for application in the tissue engineering area, particularly in tissue replacement and wound healing [131].

PAAm/cellulose nanofibers (CNF) DN gels were synthesized by simply using an alkali treatment (15 wt % NaOH) at room temperature. Investigating the morphology of this DN gel it was noticed that the CNF network was embedded in the PAAm matrix, in this manner improving the strength of these hybrid gels. The obtained PAAm/CNF DN gels present notably improved mechanical properties that are proper for application as biomedical load-bearing gel materials [126]. Hydrogels based on PVA blended with cellulose (PVA-Cel) were obtained through FT cycles and were evaluated in terms of appropriateness as a part of a structure simulating the length scale dependence of human skin [123]. CMC-PEO hydrogels and porous gel films, with excellent biocompatibility, were prepared by mixing CMC-acrylate and PEO-hexa-thiols, as precursor solutions. The porous gel films were obtained by using ammonium bicarbonate particles as porogens, prior added in the precursor solutions. The obtained hydrogels and gel films show significant potential for tissue engineering applications [141]. Hashimoto et al. [159] has fabricated an amphiphilic crosslinked porous nanogel (NanoCliP), which self assembles, and presents the ability to embedded proteins, liposomes, and cells. This NanoCliP gel was synthetized using Michael reaction, by addition of a self-assembled nanogel of acryloyl group-modified cholesterol-bearing pullulan to pentaerythritol tetra (mercaptoethyl) polyoxyethylene, followed by freezing-induced phase separation. The in vivo tests show that the NanoCliP gel brings suitable features as a scaffold for tissue engineering, demonstrating improved cell infiltration, tissue ingrowth and neovascularization as observed from Figure 3.

Zhang et al. [161] fabricated PVA-i-CG based organic hybrid hydrogels, via a facile FT technique, as tissue engineering scaffolds. The hydrogels demonstrated increased pore structure stability, enhanced attachment and proliferation of ATDC5 cells, good hemocompatibility, and low adverse effects. Li et al. [172] has prepared a DN hydrogel GG/PEGDA by combining GG with PEGDA. The effects of viscoelasticity of GG/PEGDA DN hydrogel on the biological behavior of bone mesenchymal stem cells (BMSCs) were explored in vitro and in vivo. GG/PEGDA DN hydrogel shows excellent mechanical and relaxation properties which provide a favorable physical environment for cell proliferation and spreading, and induce chondrogenic differentiation. In another study was developed a DN hydrogel based on a GG gel and a poloxamer-Hep (PoH) network (PoH/GG DNH) to overcome the drawbacks of each gel network and to enhance the microenvironment for cell delivery. The DNH system was tested on bone marrow stem cells isolated from rabbits (rBMSCs) revealing that supported cell survival, maintained cell’s morphology and phenotype. The in vivo results have demonstrated that PoH/GG DNH endorse the cell distribution, adherence, and ECM production [173].

#### 5.2.2. Wound Dressing

Gamma irradiated PVP/κ-CG based hydrogel obtained by gamma irradiation was intensively studied and applied as a biomaterial for wound dressing. This system presents several advantages such as a single step simultaneous sterilization and hydrogel formation, without the need of using initiator or crosslinker [169]. To enhance the poor mechanical strength of γ-irradiated PVA/PVP/κ-CG hydrogel, silk was added as a reinforcement agent [178]. PVP/κ-CG/PEG hydrogel dressing presents a long shelf life, have a high tensile strength, thus assuring an easy removal because it maintain its physical integrity. The advantages mentioned above make these systems to present increased patient compliance and are more effective than the commercially available ones [169]. PEG/GG hydrogel showed superior biocompatibility (N 90%), cell adhesion and improved cell growth compared to simple gellan gum hydrogel. In addition, reverse transcription polymerase chain reaction (RT-PCR) was used to confirm RPE-specific gene expression, and the result showed that it was positively influenced. As a result, it was observed that PEG/GG hydrogel promotes retinal regeneration compared to that of pure GG [171].

#### 5.2.3. Drug Delivery

Superabsorbent polymer compositions (SAPCs) based on poly(acrylic acid-co-acrylamide-co-22-acrylamido-2-methyl-1-propanesulfonic acid)-grafted nanocellulose /poly(vinyl alcohol)-P(AA-*co*-AAm-*co*-AMPS)-*g*-NC/PVA, were obtained using graft copolymerization reaction, to create a system for amoxicillin drug delivery. The SAPCs drug delivery vehicle obtained was intended to apply for the treatment of peptic and duodenal ulcers induced by *Helicobacter pylori* [122]. Smart (thermo- and pH-responsive) microgel particles based on HPC-AAc and poly(l-glutamic acid-2-hydroxyethyl methacrylate) were synthetized by emulsion polymerization. The microgel was tested for controlled delivery of insulin, being noted that the system is resistant to gastric pH (1.2) and release insulin in a controlled manner at intestinal pH (6.8) [135]. By NIPAAm/CMC copolymerization were obtained copolymeric (CP) sIPN hydrogels, which were redox crosslinked using *N*,*N′*-methylenebisacrylamide (BIS) and *N*,*N′*-bis(acryloyl)cystamine (CBA). The hydrogels were tested for egg white protein lysozyme delivery at pH 1.2 while the system cross-linked with BIS showed higher swelling and maximum release [137]. A hydrogel system based on CMC and CMPVA grafted copolymer was developed by crosslinking with adipic dihydrazide. This copolymeric hybrid hydrogel was proposed as a carrier for drug delivery and as a scaffold for tissue engineering, based on its biocompatibility with the living cells and the fact that ensures outstanding survival rate at lower polymer concentration [138]. Hydrogels based on bacterial cellulose-*g*-poly(acrylic acid) that are stimuli-responsive were fabricated using electron beam irradiation and evaluated as oral delivery system for proteins (e.g., bovine serum albumin (BSA)). This method offers the advantage that no cross-linking agents are involved, thus overcoming the eventual toxic effects related to cross-linkers use [129]. Pandey et al. [128] using microwaves irradiation has developed hydrogels based on solubilized BC/AAm as a drug delivery system for theophylline. Different sets of BC-*g*-poly(acrylic acid-coacrylamide) hydrogels were obtained through microwave-assisted graft copolymerization using NaOH/urea as solvent system. These series of hydrogels have demonstrated a pH-sensitivity, which had influence on in vitro drug release profile, namely lower level of release in simulated gastric fluid (SGF) than in simulated intestinal fluid (SIF). This behavior indicates that the hydrogels may be efficient as a potential oral, controlled-release drug delivery system for the lower gastrointestinal (GI) tract [130]. Another hydrogel based on BC-*g*-PAA was prepared by electron beam irradiation technique. BSA was loaded into the BC-*g*-PAA hydrogel and showed low release in acidic SGF and higher penetration across the intestinal mucosa. The in vivo tests revealed that the hydrogel is biocompatible and non-toxic [129]. Ceresh et al. [146] obtained copolymeric hydrogels by graft-copolymerization of acrylic acid on three types of starch (potato, corn and rice starches) via ^60^Co-gamma irradiation. The starch-based hydrogels presented potential as prolonged drug (e.g., sodium salicylate and theophylline) delivery vehicles; in Figure 4 being illustrated the rate of release of theophylline from copolymeric hydrogels.

CMC functionalized with thiol groups (obtained by reaction with cysteamine in presence of 1-(3-dimethyl aminopropyl)-3-ethylcarbodiimide hydrochloride (EDC)) was cross-linked, using dithiothreitol, with norbornene immobilized tetra-arm PEG (PEG-Nor) forming CMC-PEG hydrogels. The presence of thiol-bearing CMC into hydrogel structure determined pH sensitivity of the gels, demonstrating improved swelling and faster release of loaded BSA protein at basic pH [140]. Moreover, thermo-responsive CMC-Nor hydrogels was developed by its crosslinking with a dithiol end functionalized PNIPAAm, determining temperature-induced shrinkage of the gel, at temperatures above the lower critical phase transition temperature (LCST) (around 32 °C) [179].

Bajpai and Saxena [142] performed potassium persulfate (KPS)-initiated graft copolymerization of AAc onto soluble starch in the presence of *N*,*N′*-methylene bisacrylamide (MBA) as the cross-linker. The hydrogels obtained were pH-sensitive and enzymatically degradable, exhibiting minimum swelling in an acidic pH and extensive swelling at pH 7.4 (i.e., simulating intestinal fluid). The behavior at acidic pH is determined by the formation of a complex hydrogen-bonded structure and at intestinal pH enzymatic degradation occurred along with the swelling controlled by chain-relaxation, being suitable for colon targeted drug delivery. Saboktakin et al. [149] have obtained pH-sensitive starch hydrogels by free radical graft copolymerization of PMAA onto CMS, using bisacrylamide as a crosslinking agent (CA) and persulfate as an initiator. The pH-responsive behavior of CMS-*g*-PMAA hydrogels is characterized by a transition between the swollen and the collapsed states that occurs at high and low pH. The CMS-based hydrogels were tested for drug delivery. Double hydrophilic thermo-responsive pullulan-g-PNIPAAm copolymers with two different molecular weights of thermosensitive grafts were synthesized and used for preparation of indomethacin-loaded nanoparticles by dialysis and nanoprecipitation method [180]. The sustained-release properties of poloxamer 407-based in situ gel were enhanced by the combination with CG, and present high potential to be used in vaginal in situ gel drug delivery systems with prolonged local residence and therefore for better clinical outcome [160]. Hamcerencu et al. [176] performed free radical grafting/polymerization of unsaturated esters (gellan maleate) with NIPAAm, using *N*,*N′*-methylenebisacrylamide as cross-linker, to design thermosensitive hydrogels. These hybrid hydrogels were tested for their swellability, in vitro loading and release of different drugs (e.g., adrenaline and chloramphenicol) and in vivo biocompatibility. By in vivo evaluation was not observed necrosis, calcification and acute inflammation, only the formation of a thin fibrous capsule around the implanted hydrogels, thus they being preliminary proposed for ophthalmic applications.

#### 5.2.4. Other Biomedical Applications

The BC/PGA hydrogels were prepared by ^60^Co γ-irradiation crosslinking method. The BC nanofibers and PGA can form the multicomponent hydrogels with double crosslinking structure via γ-irradiation. The addition of BC increases compressive strength, storage modulus (G’) and the gel fraction but decreases the equilibrium swelling ratio of the BC/PGA composite hydrogels. The compressive strength and storage modulus of hydrogels increase 5 times and 10 times respectively at the irradiation dose of 50 kGy. Moreover, the BC/PGA hydrogels are non-toxic, indicating their safety for biomedical application [132]. By UV photo-crosslinking were obtained temperature sensitive hydrogels based on hemicellulose (Hce) obtained from acetic acid pulping of Eucalyptus and NIPAAm. The protocol involved two steps; firstly, a Hce derivative was synthetized by grafting MA to Hce that contains vinyl bonds within the side chains followed by UV photocrosslinking of Hce-MA with NIPAAm in LiCl/DMF solvent. The equilibrium swelling ratio and morphology of the hydrogels were dependent on environment temperature, implying their potential as smart materials for medical application [134]. All-trans retinoic acid aqueous gels composed of ι-CG and polyethylene oxide were proposed to be applied as a topical treatment of skin. In these gels, the PEO was selected for its high mucoadhesion property and spinnability, while ι-CG was chosed for its texture modification property and gelling feature. Combination of these components maximizes the optima properties of each entity by reducing the drawbacks of each individual polymer [162].

Deng et al. developed a novel κ-CG/PAAm (KC/PAAm) DN hydrogel through a dual physical-crosslinking strategy, with the ductile, hydrophobically associated PAAm being the first network, and the rigid potassium ion (K^+^) cross-linked KC being the second network. The DN (DPC-DN) hydrogels with optimized KC concentration exhibit excellent fracture tensile stress and toughness, comparable to those fully chemically linked DN hydrogels and physically-chemically cross-linked hybrid DN hydrogels. Additionally, DPC-DN demonstrated rapid self-recovery, remarkable notch-insensitivity, self-healing capability, as well as excellent cytocompatibility towards stem cells [167]. In a similar manner were obtained hybrid hydrogels based on Iota-Carrageenan and polyacrylamide to be used as matrix for silver nanoparticles designed for bacterial inactivation applications [181].

## 6. Heteropolysaccharides-Based Hybrid Hydrogels

### 6.1. Ability of Heteropolysaccharides to Form Hybrid Hydrogels

A biocompatible and biodegradable heteropolysaccharide that forms hydrogels by mixing with multivalent cations is the alginic acid [182,183]. Spherical core–shell gel-bead structures (or worms) were obtained by combining alginic acid with 1,3,2,4-di-(4-acylhydrazide)-benzylidenesorbitol (DBS-CONHNH_2_) [184]. The gels based on alginic acid proved to have important applications in domains like drug delivery and tissue engineering [185].

Microspheres (MS) of hybrid hydrogels that can adjust their mechanical properties and durability in function of the biological environment were obtained using SA with heterotelechelic PEG derivatives [186]. These hydrogels are appropriate for cell transplantation applications.

For enhancing the ability of liquid uptake and the mechanical properties of the hydrogels based on chitosan (CS), this natural polymer was associated with synthetic polymers or grafted with vinyl monomers, such as acrylic acid and acrylamide [187,188].

Chen et al. [189] prepared macroporous PVA/CS hydrogel sponges that showed higher antimicrobial and haemostatic activity than pure CS sponges.

Hyaluronic acid (HA) is abundant in connective, epithelial, and neural tissues [190]. HA macromolecules showed anti-inflammatory, immunosuppressive properties and block angiogenesis, while cleaved small fragments induce opposite behavior, enabling endothelial cells migration and angiogenesis [191]. Kim et al. [192] obtained PVA/HA hydrogel nanofibers by chemical crosslinking, using HCl and glutaraldehyde. They observed that the swelling ratio of these hydrogels is higher in respect with that corresponding to pure PVA hydrogel. A good biocompatibility of PVA/HA hydrogel nanofibers was evidenced by a higher cell adhesion at their surfaces, independent on the HA presence.

Heparin (Hep) has a high negative charge, the 3-D hydrogels based on it being used in tissue engineering, implantation, biosensor domain, drug delivery. Because Hep poses some safety problems (because it is often obtained from animal sources), analogous Hep-mimicking polymers and hydrogels obtained from synthetic sources were proposed.

The use of Hep in hydrogels by delivery growth-factors generates proliferation signals to cells because of its protein polysaccharide interactions closely mimicking the native structure and functioning of ECM [193].

Supramolecular hybrid hydrogels self-assembled were obtained from low-molecular-weight gelator (LMWG, which are small organic molecules which self-assemble in water or organic solvents, forming a 3D network that entraps the liquid phase resulting in gel formation) building blocks with the polymer gelator (PG) (e.g., calcium alginate) [184]. This type of hydrogel can be used in regenerative medicine [194], in controlled drug delivery [195], or in electronics devices as patterned conducting gels where they contact interface with living media. The components usually used for obtaining self-assembled multi-component hybrid hydrogels are: a pH activated LMWG, a temperature activated PG, an anionic biopolymer (such as Hep) and a cationic system capable of binding Hep.

### 6.2. Biomedical Applications of Heteropolysaccharides-Based Hybrid Hydrogels

Some examples of heteropolysaccharide-based hybrid hydrogels used in different biomedical domains are listed in the Table 3 where are also mentioned preparation methods and general properties.

#### 6.2.1. Tissue Engineering

Generally, the scaffolds used in tissue engineering should have several properties, such as biocompatibility, cell proliferation, controlled swelling, ease of administration, antimicrobial, stability, porosity, adhesion, low immunogenicity, colonization of host cells without inducing any histological changes, integration with host tissues [249,250,251,252], biodegradability, bio mineralization, non-toxic degradation products, and also degradation of scaffolds should be inversely proportional to the rate of synthesis of the newly regenerated tissue [253].

Pok et al. [254] obtained 3D scaffolds of self-assembled PCL in a gelatin-CS hydrogel, for possible application in congenital heart defects. They observed similarities between the mechanical properties of the hydrogel with those of the native tissue, as well as migration of neonatal rat ventricular myocytes (NRVMs) [254]. Zhao et al. [228] synthesized hydrogel scaffolds by chemical crosslinking between quaternized CS and polyaniline, using oxidized dextran as cross-linker. The obtained hydrogels presented decreased cytotoxicity, higher antibacterial activity, and enhanced proliferation of C2C12 myoblast cells when compared with quaternized CS hydrogel. These hydrogels could be used for muscle, nerve, and cardiovascular repair [228]. PVA hydrogel was loaded on one side only with Hep for possible application in vascular tissue engineering [255], because release of Hep from PVA/Hep hydrogel can prevent clot formation.

HA-based hydrogels are often used in cartilage tissue engineering, because HA has an inhibitory effect on fibronectin fragment-mediated chondrocytic chondrolysis [256], inhibitory effects on prostaglandin synthesis, proteoglycan release [257], and degradation by enzymes and free radicals [258]. One of the main disadvantages of using HA in cartilage tissue engineering is its poor mechanical properties. That is why, this natural polymer has to be used together with various synthetic polymers, such as PNIPAAm and PEG [259].

Fan et al. [260] evaluated the potential of hybrid poly(lactic-*co*-glycolic acid)-gelatin/chondroitin/hyaluronate (PLGA-GCH) scaffolds in cartilage repair. It was observed that differentiated mesenchymal stem cells (MSCs) seeded on PLGA-GCH significantly increased the proliferation of MSCs and GAG synthesis compared with PLGA scaffolds.

Bichara et al. [261] developed a flexible PVA/SA hydrogel, using human nasal septum chondrocyte cells; the systems have been implanted into the subcutaneous environment of nude mice. In vivo tests showed deposition of collagen type II in the hydrogels, behavior that recommend this hydrogel type for reconstruction of craniofacial cartilage.

Kunisch et al. [262] prepared star PEG/Hep hydrogels trying to prevent mineralization of the upper cartilage zone, for inhibiting long-term progression of calcified cartilage into bone.

A thermo-sensitive copolymer hydrogel was obtained by grafting PNIPAAm onto HA. This system passed from a liquid-like behavior to an elastic gel-similar one, at 30 °C, this fact being useful for cell encapsulation in the hydrogel [263]. Another thermo-sensitive hydrogel was prepared using Pluronic and HA, this one being a potential candidate for applications as artificial vitreous substitute [264].

Self-healing hydrogels can be successfully used in drug/cell delivery or in 3D printing [265]. Self-healing hydrogels based on glycol chitosan and difunctionalized PEG (GC-DP) were also used for tissue repairs, in central nervous system [266], or for inducing blood capillary formation. In order to achieve this second purpose, a multicomponent hybrid hydrogel was obtained using an IPN of GC-DP and fibrin [267]. The hydrogel induced vascular endothelial cells to form capillary-like structures; injection of this hydrogel promoted angiogenesis in zebrafish and rescued the blood circulation in ischemic hindlimbs of mice.

Fares et al. prepared IPN and sIPN hydrogels, using a pectin grafted polycaprolactone (pectin-*g*-PCL) and a gelatin methacryloyl (GelMA) component [268]. The IPN hydrogels were characterized by cytocompatibility and, in the meantime, induced the growth of MC3T3-E1 preosteoblasts in vitro, proving that they are appropriate for different applications in tissue engineering.

A pectin-Fe^3+^/polyacrylamide hybrid DN hydrogel was developed by Niu et al. [269]. These hybrid DN hydrogels were characterized by very good mechanical properties (such as stiffness, fatigue resistance, notch-insensitivity), as well as a high-water absorption ability (85%). All these characteristics recommend this hydrogel type to be used in the load-bearing tissue repair field.

Injectable scaffolds are superior to preformed scaffolds in terms of improved patient’s compliance, ease of clinical implementation for the treatment of geometrically complex, and large lesions via minimally invasive techniques, such as arthroscopy [270]. This type of scaffolds can be used in minimally invasive surgical procedures; they completely fill the defect area and have good permeability, being hence promising biomaterials [271,272]. The technique can be effectively applied to deliver a wide range of bioactive agents, such as drugs, proteins, growth factors, and even living cells. For the development of such type of scaffolds, natural polymers were used (i.e., collagen, chitosan, gelatin, alginate, hyaluronan, chondroitin sulfate, pectin) [273]. In order to obtain in situ gelling systems, different techniques can be applied, such as photo-crosslinking, chemical crosslinking, enzymatic crosslinking, pH-induced gelation, temperature-induced gelation, ionic and hydrophobic interactions [259].

#### 6.2.2. Wound Dressing

Due to its properties, CS proved to be an attractive candidate for treating wounds, even major burns [274,275].

PVA/HA membrane hydrogels were tested for wound dressing application, from the point of view of their biological properties and biocompatibility. Increasing the HA content in the hybrid hydrogels, a decreased migration and cell viability were observed, due to an increase in the viscosity of the PVA/HA system. In the absence of ampicillin, the obtained membrane hydrogels were active against *Candida albicans,* while when ampicillin was added, they proved to be effective also against *Staphylococcus aureus*, but not against *Escherichia coli* [276]. HA/PVPA/CS hydrogel designed for being used for skin wound healing showed antimicrobial activity against *E. coli* [277].

#### 6.2.3. Drug Delivery

Hydrogels represent a drug delivery system class that has excelled as smart drug delivery [105,278]. Biocompatible, biodegradable hydrogels have been designed using natural polymers that are susceptible to enzymatic degradation, or using synthetic polymers that possess hydrolysable moieties. CS positive features e.g., hydrophilicity, functional amino groups, and a net cationic charge recommend its hydrogels for the intelligent drug delivery and of macromolecular compounds, such as peptides, proteins, antigens, oligonucleotides, and genes [220,279].

Wu et al. [280] developed hydrogel-based N-[(2-hydroxy-3-trimethylammonium) propyl] chitosan chloride (HTCC) and PEG for insulin release. Hydrogen bonds among amino groups present in insulin and hydroxyl groups present in PEG or HTCC allowed prolonged drug release. After spraying of formulation into nasal cavity, the solution formed gel at body temperature. This hydrogel system presented lower mucosal clearance and sustained in site targeted drug release. The results showed that the hydrogel can be used as nasal delivery carrier for protein or peptide drugs [280].

When oral drug delivery is not practicable, nasal administration of the CS hydrogels can be used for delivery of peptides and vaccines [281].

CS/PEG hydrogels were tested for drug delivery at the level of the gastrointestinal tract [282]. The release rate of the drug from the PE/CS systems was delayed when compared with those only with CS [283], maybe because PEG can improve the CS solubility [284], thus increasing its transfection efficiency when used as a gene carrier [285].

IPNs obtained from CS and PEO proved to be appropriate carriers for drug delivery systems specific for the stomach, being effective for *Helicobacter pylori* treatment [286].

For colon drug delivery, CS-polyacrylic acid (PAA) hybrid hydrogels were tested, these ones being biodegraded by colonic normal flora [287].

CS/PVA hybrid hydrogels were studied as potential systems for drug delivery, using as drug model: PTX, insulin, BSA; the properties of these hydrogels (such as antitumoral activity, period of drug release) were superior to those of other delivery systems [209,212].

Zhou et al. [288] developed a CS/poly(oligo ethylene glycol) system, used as a controlled drug release in the chemo-cryo cancer therapy.

Åhlén, Tummala, and Mihranyan [289] reported that contact lenses based on CS-PAA nanoparticles and PVA hybrid hydrogels had greater potential for extended release during 28 h.

Due to the highly negative effects of the usual chemotherapy treatments, polysaccharide hydrogels were tested as drug carriers, in order to obtain a controlled/localized drug release. As examples are the 3D CS/PVA hybrid hydrogels developed by Jamal et al., which described the great potential in inhibiting angiogenesis of these hybrid systems [290].

Islam and Yasin [291] developed CS/PVA porous hybrid hydrogels crosslinked with tetraethoxysilane as a drug delivery system for dexamethasone. By increasing the PVA concentration it was obtained a decrease of the swelling degree of the hydrogels. The pH media also affected the swelling degree, the minimum swelling being observed in acidic and basic media, and the maximum around a neutral pH. Hydrogels released around 9.4% dexamethasone during the first two hours, the released amount increasing up to six hours (Figure 5).

Yang et al. [292] developed GC and DP for intra-tumoral therapy in vivo. GC-DP hydrogel containing antitumoral drug was injected into the disease site, for being released in situ. Moreover, the ionic GC-DP hydrogel exhibited microwave susceptibility to produce high-temperature hyperthermia for tumor ablation [293].

A thermoresponsive nano-sized chitosan-grafted PNIPAAm (CS-*g*-pN) hybrid hydrogels curcumin-loaded was developed as an advanced material that can be functionalized and optimized for targeted therapy and controlled delivery of small molecules and/or biomolecules [294].

For treating oral mucosa ulcer, one can use antibiotics, analgesics, adrenocortical hormones, and glucocorticoids, drugs that may induce undesirable side effects. That is why, Luo et al. prepared four different injectable CS based thermogels, using PNIPAAm and PAAm, synthesized by an in situ free radical polymerization procedure. Hybrid hydrogelswere tested from the point of view of antibacterial activity against Gram-negative (*Escherichia coli*) and Gram-positive (*Staphylococcus aureus*) bacteria, human gingival fibroblasts viability and growth, therapeutic effect, hemostatic activity [295].

All the four CS-based hydrogels proved antibacterial activity, the inhibition rate of the studied bacteria significantly increasing with increasing hydrogel concentration up to 5 wt%. The antibacterial activity was higher for the samples with a higher CS content (Figure 6a,b). In the meantime, all the four CS-based hybrid hydrogels induced no important toxicity towards human gingival fibroblasts and were characterized by good hemostatic properties. By comparison, the samples with the highest CS and PNIPAAm content (samples 1 and 2) were more effective in treating oral mucosa ulcer. Another important feature of some of them (i.e., 1, 2, and 4, see Figure 6) can reversibly form semi-solid gels at physiological temperature, being easily applied to oral cavity by injection.

The in vitro release of the heat shock protein 27 (HSP27) (protein that protects heart muscle for ischemic injuria) was released over a period of 14 days from a hybrid hydrogel PLGA/Alg containing also TAT peptide [296]. After injection of this system in a myocardial infarction model, some parameters describing the heart state were significantly improved.

SA/NIPAAm hydrogels, chemically crosslinked with *N*,*N*’-methylene bis-(acrylamide), that respond at the fluctuations in temperature and pH proved to be suitable for sustained drug release of paracetamol and theophylline [297], a better drug entrapment and a slower drug release being obtained.

Thermo-responsive hydrogels for injectable drug administration in chemotherapeutic treatment were reported by Chen et al., who prepared hexamethylene diisocyanate (HDI)-pluronic F 127 copolymer /HA systems [298]. Injectable hydrogels were also prepared by conjugating HA with two types of complementary single-stranded DNA (HA-DNAs) [299].

Both bFGF and vascular endothelial growth factor (VEGF) were entrapped in PVA/Hep hydrogels [300]. When compared with PVA gels, the hybrid PVA-Hep gels induced a significantly lower fraction of initial release of bFGF in the first 12 h, as well as a significantly decreased quantity of released bFGF, behavior observed for the entire followed period of time (Figure 7a). The same behavior was also evidenced when VEGF was encapsulated in PVA-Hep gels (Figure 7b). This fact suggests that PVA-Hep gels are appropriate for being used in controlled-drug release applications. In what is concerning the dual release of bFGF and VEGF, a synergistic effect was observed (Figure 7c).

Hep-based injectable hydrogels can be used to mimic the extracellular matrix as promising drug delivery systems for postoperative chemotherapy, cell delivery carrier and the regeneration of damaged liver or other tissues. Hep-based nano-hydrogels are commonly applied for cancer cell-targeted delivery, as carriers for anti-fibrotic and anti-cancer agents and gene delivery [301]. Since Hep has high cost, dramatic loss in bioactivity and degradation when using covalent or non-covalent strategies for obtaining the hydrogels, interference with blood components), analogous Hep-mimicking (also called Hep-inspired) polymers are one of the current research hotspots to substitute the usage of Hep in the fabrication of hydrogels. Sulfonated polymers and sulfated glycosaminoglycan have been widely recognized as Hep-inspired components since they show similar bioactivity properties as Hep, such as anticlotting and antithrombotic activities, stabilization of growth factors, and promotion of angiogenesis. Between the main applications of the Hep-inspired hydrogels, one can mention: cell culture, loading of drugs/molecules, blood contacting applications.

## 7. Hybrid Proteins Based Hydrogels for Biomedical Applications

### 7.1. Ability of Proteins/Peptides to Form Hybrid Hydrogels

Protein-based materials are popular as engineering bioactive scaffolds because of their advantages in mimicking the extracellular environment [302]. Hydrogels-based on proteins are applied in biomedicine field as tissue engineering materials, drug delivery, etc., because they are easily degraded by the body and display a high biocompatibility.

Proteins can be employed as building blocks, systems embedded with particles, etc. In the design of most hybrid hydrogels composed with synthetic polymeric, peptides act as structural elements. They are generally used in hybrid systems for the improvement of mechanical properties of other polymers in the sense of malleability, and also to induce biodegradability and biocompatibility. Variety of functional groups belonging to proteins/peptides allows the physical cross-linking through hydrogen bonds, electric interactions, and/or π-π stacking.

Additionally, chemical crosslinked hybrid hydrogels can be formulated using protein—synthetic polymer couple. Common cross-linkers such as glutaraldehyde, formaldehyde and carbodiimide, have been widely used in the fabrication of protein-based hybrid hydrogels in the past decades [303]. Tetrakis (hydroxymethyl) phosphonium chloride (THPC), possessing four hydroxymethyl arms, is an effective, mild and low-cost cross-linker to make protein hydrogels. Various natural proteins, including BSA, gelatin, silk fibroin, milk protein, soy isolate protein, ovalbumin and lysozyme, could be cross-linked by THPC to form high strength hybrid hydrogels with rapid self-recovery properties [304]. Biocompatible protein crosslinkers of proanthocyanidin, a kind of naturally occurring polyphenol extracted mainly from plants [305] and genipin [306] as natural crosslinkers that can be used to obtain biocompatible hybrid hydrogels based on proteins. Multi-functional natural proteins macromonomers can be also used alone as macro-crosslinkers, eliminating the need for a conventional crosslinkers.

### 7.2. Properties of Proteins to Form Hybrid Hydrogels for Biomedical Applications

#### 7.2.1. Collagen

Collagen an important insoluble protein of the human and of animal body is met in skin, connective tissue, cartilage and bones. Of all twenty-nine types of collagen that have been discovered, type I is frequently used as a component in the development of biomaterials. Collagen had gained a considerable reputation in the biomedical field due to its unique properties such as low immunogenicity, low toxicity, excellent biocompatibility, good safety, biodegradability, weak antigenicity and ability of skin, bone, or other tissue regeneration [307]. From the structural point of view, collagen based hybrid hydrogels can mimic the physiological ECM, stimulating cell migration, proliferation and adhesion, and providing bio-safe profiles and no chronic inflammatory response. Collagen hybrid hydrogels have been applied for the partial or the whole reconstruction and healing of different parts of human body including skin tissue [308], bone, cartilage [302], blood vessels, cornea [309], and brain parts [310].

#### 7.2.2. Gelatin

Gelatin is ussualy extracted from animal (bovine, porcine) and fish (jelly fish, sea urchin) sources by acid (type A gelatin) or basic (type B gelatin) hydrolysis of collagen [311]. The physically gelatin gels are thermo-reversible, thus during cooling, the random coil structure is in part rebuilded to a triple helical structure [312]. The isoelectric point of type A gelatin varies between pH 7 and 9, while in case of type B gelatin, the isoelectric point is placed from pH 4.7 to 5.4. Cationic or anionic charged gelatin is usefful for loading therapeutic principles (molecules, drugs) through electrostatic interactions. In contrast with collagen, gelatin manifests good stability at high temperature in a broad interval of pH [313]. Synthetic polymers such as PEG and PVA are some of the most used for the obtaining of hybrid gelatin based hydrogels with application in almost all the medical domains.

#### 7.2.3. Keratin

Keratin is a group of fibrous proteins that forms the bulk of cytoplasmic epithelia and epidermal structures and is abundant in hair, nails, wool, horns and feathers. Studies on keratin-based biomaterials are mainly focused on keratin extracted from wool and poultry feathers, but more especially from human hairs because being human-derived, the risk of immune response is reduced [314]. Two conformations can be found in keratin, α-helix and β-sheet. Like other naturally derived protein biomaterials, keratin possesses 14 types of amino acids, where cysteine plays an important role in the formation of disulfide bonds that influence the high mechanical strength of keratin.

When extracted reductively [315], the resulting material is known as kerateine (KTN) with thiol groups able of forming disulfide bonds. When extracted by oxidative means [316], is referred to as keratose (KOS). The thiol groups of the cysteine residues in KOS are “capped” as sulfonic acid residues and are unable to form disulfide bonds. These chemical differences in the proteins are known to affect the physical properties of biomaterials, particularly hydrogels, derived from keratins. KOS will form hydrogels through physical entanglements, but degrades rapidly due to the lack of covalent disulfide crosslinks. KTN persists much longer due to the presence of both physical entanglements and covalent disulfide crosslinks [317]. Unlike collagen, keratin-based hydrogels can stay stable and resistant to biodegradation in vivo for a longer period without being degraded by enzymes because there is no keratinase or other keratin-degrading enzymes in humans and animals bodies.

In the form of hydrogels, keratin scaffolds have porous gel walls and large voids which are suitable for the cells proliferation. When keratin is combined with synthetic polymers, generally keratin-based hybrid hydrogels are obtained with potentially applications in drug delivery systems or wound healing [318].

#### 7.2.4. Bovine Serum Albumin

Bovine serum albumin (BSA) is frequently used as an active principle, but in some cases as a component in a hydrogel system. because its low cost, stability, specific ligand-binding properties and increased solubility [319].

#### 7.2.5. Silk

Silk, a fiber protein produced by the silkworms, is composed of two main proteins called silk sericin (SS) (25% of the total weight of raw silk) and silk fibroin (SF) (75%) [320]. Sericins, amorphous and hydrophilic proteins, act as a gummy/adhesive substance that joins the fibroin filaments. This glue-like sericin protein gets wrap around the SF, what is hydrophobic, highly crystalline with an oriented structure [321]. Silk based biomaterials has various advantageous features, including excellent biocompatibility, controllable biodegradation, and desirable mechanical properties, are nontoxic, nonimmunogenic, and have been approved by the United States Food and Drug Administration (US FDA) for use in the human body for sustained-release drug delivery systems, bone and skin tissue regeneration and repair, biosensor, and 3D bioprinting [322]. It was demonstrated that SS plays a crucial role in the antibacterial process of wound treatment. The sol–gel transition of aqueous solution of SF is a natural process which without an exterior stimulus is quite long, usually a week to a month, which may limit its practical use [323]. Therefore, there are some factors that can stimulate and enhance the gelation kinetics of SF. For example, reducing the electrostatic repulsion by lowering the pH (<5), increasing the gelation temperature (>60 °C), increasing in protein sol-gel critical concentration (> 5–10% (*w*/*v*)), blending with polyhydric alcohol agents, or adding divalent ions (Ca^2+^) can decrease the gelation time by attenuating the hydrophobic interactions amongst protein chains [324]. Physical methods such as ultrasonication, or chemical crosslinking or a polyreaction of additive micromolecular agents, accelerate the gelation of fibroin. Nevertheless, these physically or chemically stimulated sol–gel transitions are almost irreversible. During the gelation process fibroin molecules rearrange from a random coil conformation in the sol state to an antiparallel β sheet conformation in the gelled state [325]. The presence of a large amount of Gly-Ala repeats units in fibroin favors the formation of β sheets which accelerate its gelation. Silk polymers can be engineered to produce thermo-sensitive hydrogels. At room temperature, the thermo-sensitive hydrogel may remain liquid but at body temperature exists as a hydrogel.

#### 7.2.6. Resilin

Resilin is a pliable and extendable structural protein found in insects. It is named for its resilience to repeated rounds of stretching and relaxation [326]. Resilin has ability to store mechanical energy [327].

#### 7.2.7. Whey Proteins

Whey proteins (WP) are globular milk-derived proteins and contain as major protein fractions β-lactoglobulin, α-lactalbumin and BSA. They are extremely inexpensive and abundantly available in various forms (concentrates—WPC, hydrolysates—WPH, and isolates—WPI). Heat treatment of an aqueous solution of whey protein isolate (WPI) above 60 °C results in its unfolding followed by the formation of new inter-and intra-protein bonds that create a three-dimensional gel network [328]. Biodegradability and ability of WPI to form a hydrogel without the use of chemical cross-linking agents makes it attractive for use in biomedical applications.

#### 7.2.8. Soy Protein Isolate

Soy protein isolate (SPI), contains two major components: glycinin (52%) and conglycinin (35%), with hydrophobic components in the molecular structure [329]. Soy protein has advantages over the various types of natural proteins employed for biomedical applications due to its low price, nonanimal origin, and relatively long storage time and stability. Because of its globular structure, soy protein is more resistant to hydrolysis compared with coiled or helical structures. SPI is an electroactive protein with an increased content of polar amino acid moeties that produces charges in various conditions of pHs especially in strong acidic or basic pH [330], which recommend its to be used as a natural protein-based electroactive hydrogel for microsensor and actuator, particularly in the biomedical area. Due to their flexibility, SP materials could also be successfully processed into scaffolds by a 3D printing [331]. Residues resulted after the degradation process of SP based hydrogels were demonstrated to be non-toxic and also are capable to promote collagen deposition in cultures of fibroblast cells and to determine the mineralization in the presence of osteoblasts, hence sustaining the ideea that SP manifest intrinsic bioactivity [332].

Peptides derived from native soy protein may be employed as excellent building blocks to fabricate hybrid hydrogels [333]. The procedure includes the dissolving of native SPI in alkaline solution (such us urea) where it is denatured and unfolded to peptide chains. Thus, the unfolded peptide chains will induce an increase of protein solubility in water. Moreover, the exposed sulfhydryl and hydrophobic groups on the peptide chains will be employed for further chemical reactions. The use of SP chains as structural components into hydrogels have some advantages such as cell and growth factor or surface binding and electroactive characteristics [13].

Due to their various potential biomedical applications, hydrogels based on engineered proteins have attracted considerable interest [334]. Calmodulin (CaM) is a calcium-binding protein with an important role in the biological recognition being used as an element in stimuli-sensitive hydrogels. It has the ability to manifests a large conformational change on binding calcium, certain peptides, and the phenothiazine group of drugs (anti-psychotics). CaM undergoes two different types of conformational changes, an apo-state in the absence of a ligand, and a holo-state when it is bounded to a ligand [335]. Thus, in the presence of Ca^2+^, CaM undergoes a rapid transition from an extended dumbbell conformation to a collapsed (more constrictive) conformation in response to binding of ligands (small molecule drugs, peptides, a variety of proteins). When Ca^2+^ is removed from CaM, the protein changes from its bound conformation to native conformation. New classes of CaM based hydrogels should lead to a new breed of intelligent biomaterials that could find many applications in the field of responsive drug delivery systems, as well as in a variety of microfluidics systems and BioMEMS devices [22].

#### 7.2.9. Elastin

Elastin represents a structural protein of the ECM providing tensile strength and elasticity. Natural elastin hardly has been used as a hydrogel in biomaterial field. An important requirement when using proteins as biomaterial is the purity. During the synthesis, elastin can be contaminated and may induce immunological responses. Additionally, elastin is insoluble and has a strong tendency to calcify, making purification even more difficult [336]. Consequently, soluble forms of elastin including tropoelastin [337], α-elastin [338], and elastin-like polypeptides (ELPs) [339] are frequently used to form crosslinked hydrogels.

The gelation process of ELPs is influenced by the temperature. When the temperature is reduced under a certain value, the hydrophobic groups are fenced by ordered water molecules from hydrophobic hydration, and ELPs became soluble. Above this transition temperature, the H_2_O molecules surrounding the ELPs become bulky being less-ordered, and thus the protein is collapsed. This leads to the folding and self-assembly of ELP, and consequently the gelation occurs [13,340]. No hydrogels based on elastin and synthetic polymers have been reported in literature.

Self-assembled Leucine zipper (LZ) was first studied by Petka et al. started with 1998 [341]. From that moment, LZ building blocks were involved in the devlopment of self-assembled protein based hydrogels, initiating remarkable possibility for using genetic engineering proteins to adjust the physical and functional properties of hybrid hydrogel materials [342]. The high flexibility of the random-coil like polypeptides incline to form intramolecular loops, conducting to an unwanted and accelerated erosion rate of the hydrogel matrix [343]. Peptides with different biological functions can be incorporated in the LZ protein backbone to create a functional chimeric protein or fusion proteins with self-assembling properties [344] for the use in tissue engineering applications. Shu et al. designed polypeptide-polymer conjugates using PEG chains covalently bounded to the exterior side chain of peptides forming tertiary structures, that could be leucine zipper [345]. This knowledge is usefull for the development of new self-assembled and responsive protein based hybrid hydrogels.

### 7.3. Biomedical Applications of Protein Based Hybrid Hydrogels

The main biomedical applications of the protein based hybrid hydrogels are summarized in Table 4 Some of the most recent (from the last 6 years) application in the biomedical field of protein based hydrogels in combination with a synthetic polymer are given in Table 5. As can be observed, most of the researchers are trying to find the most biocompatible way to develop a new biomaterial without the use of chemical cross-linkers (which are very toxics) even if the gelation process is formed by chemical or physical interactions.

#### 7.3.1. Tissue Engineering

In situ forming hydrogel systems have attracted considerable interest as injectable scaffolds for tissue engineering and drug delivery due to their easy applications and minimally invasive injection procedure. By modification of gelatin with hydroxyphenyl propionic acid (HPA) and conjugation of 4-arm-polypropylene oxide−polyethylene oxide (4-Arm-PPO-PEO = Tetronic) with tyramines, Park et al. succeeded to prepare an injectable hydrogel by mixing the solutions of the two components in the presence of horseradish peroxidase (HRP) and hydrogen peroxide (H_2_O_2_) [379].

During the enzymatic coupling reaction under physiological conditions, HRP facilitated to the phenol molecules from both in modified gelatin and conjugate, to participate to the coupling reaction by interaction via C-C bond in ortho position or with C-O bonds at phenoxy oxygen. After the subcutaneous injection on mice, gelatin based hybrid hydrogel was quickly created allowing the natural tissues growth into the hydrogel network.

SF has also been considered as a candidate for 3D bioprinting, as the protein polymer chains can be physically crosslinked through intermolecular and intramolecular β-sheet structure formation via hydrophobic interactions to stabilize the materials without the need for chemical reactions or additives. Thus, SF/PEG hydrogels were studied as self-standing bioinks (biological ink) for 3D printing use in tissue engineering [380]. Mixing PEG with silk induces physical crosslinking and thus rapid gelation of silk and water insolubility. After subcutaneously implantation in mice, the bioink gel with and without fibroblast maintained shape and structure after 6 weeks, and a significant amount of cells remained alive in the gel matrix.

One single example of hybrid resilin based hydrogel have been reported in literature by McGann et al. that was composed of a resilin-like polypeptides (RLPs) and a multi-arm PEG macromer [381]. This hybrid hydrogel can be rapidly cross-linked through a Michael-type addition reaction between the thiols of cysteine residues on the RLP and vinyl sulfone groups on the multi-arm PEG. The obtained elastic and resilient hydrogels are capable to encapsulate human mesenchymal stem cells (hMSCs), are biodegradable and possess rubber-like properties that would be useful for mechanically-demanding tissue engineering applications, especially those aiming to remedy cardiovascular pathologies.

Liu et al. obtained photo-crosslinkable physical hydrogels with practicability in artificial ECM. Before the hydrogel synthesis step, this research group created macromers comprising a hydrophilic chain with a terminal self-assembling leucine zipper domain A and a terminal photoreactive acrylate group (PEGDA) [382]. LZ domain A was chosen due to its ability to undergoes tetrameric physical association that will allow the hydrophilic polymers to self-assemble into four-arm macromers. Then, the four-arm macromers were photo-crosslinked into hydrogels, where their biodegradability and ability to mimic nonproteolytically mediated cell migration and outgrowth by reversible opening and closing the 3D cell migration paths in biological systems, was demonstrated.

#### 7.3.2. Bone Tissue Engineering

Sometimes, the injectable property of traditional protein based hydrogel is unsatisfactory, which cannot provide adaptable filling of lesion defects with irregular shapes. Thus, the adding to it of a synthetic polymer with propper properties can be the solution for these problems.

There are hydrogels that are not suitable for applications in musculoskeletal systems, because most of them often exhibit too weak mechanical properties owing to a decreased elastic modulus ranging from kPa to MPa, in comparison with the native bone tissue that exert a modulus between 1 and 20 GPa. Because human cells respond differently to various mechanical stresses, such as compression, tension and shear [383], it is essential for a hydrogel scaffold to support loads and movements when is used an implant. Fortunately, by growing the number density of crosslinks and the polymer concentration inside the network gel or by combining two or more IPNs, this way has allowed scientists to obtain hydrogels with increased stiffness. Microfabrication techniques such as electrospinning and 3D printing have also proven to be successfully used in the designing of hydrogels with strong and complex structures for bone engineering applications [384].

Liu et al. [385] obtained a biomimetic bone substitute comprising a type I collagen matrix gel in which poly(l-lactide-*co*-caprolactone) nanoyarns were incorporated before gelation using a water vortex as collector, instead of traditional rotating drums or dual metal collection rings, to produce aligned nanoyarns in order to increase the mechanical strength of the resulted hydrogels allowing in the same time the cell proliferative ability of collagen. The nanoyarns were short enough to eliminate the entanglements formation when they were suspended in the protein solution.

Gan et al. [386] studied the interpenetration between a primary network made of dextran and gelatin and a secondary network composed of PEG. The resulted hybrid hydrogels exhibited improved toughness and good proliferation, clustering and adhesion of the incorporated nucleus pulposus cells, when the proportion of the natural network was 4-fold greater than the synthetic one.

#### 7.3.3. Cartilage Tissue Engineering

Several hybrid hydrogels have been developed as injectable scaffolds to mimic the ECM of cartilage. Wang et al. [387] obtained in situ injectable silk solutions of mixed silk and low-molecular-weight PEG formed hydrogels in less than 30 min when the concentration of PEG in the gel was 40–45%. A scaffold/nano- or micro-particles system is introduced for accelerated healing by sustained release of drugs/biomolecules or/and by rapid cell proliferation. Thus, gelatin can be part of the scaffold or of the particles. For example, Xu et al. reported the fabrication of the alginate-gelatin hydrogel microspheres using an electrospray technique, that were embedded with human bone marrow stromal cell (hBMSCs), and then seeded and assembled in 3D-printed PCL scaffolds for the fabrication of a mechanically stable and biologically supportive tissue engineering cartilage construct [388]. In Asadi et al. study [389], gelatin was used as a component of hydrogel scaffold. In this case, PCL−PEG−PCL nanoparticles loaded with transforming growth factor β1 (TGFβ1) were embedded in the gelatin hydrogel scaffolds and investigated as system for cartilage tissue engineering.

Lee et al. developed a strategy to engineer an auricular cartilage using SF and PVA hydrogel [390]. They demonstrated that an intact 3D ear-shaped auricular cartilage formed six weeks after the subcutaneous implantation of a chondrocyte-seeded 3D ear-shaped P50/S50 hydrogel in rats.

#### 7.3.4. Wound Healing

An ideal scaffold for skin tissue regeneration is expected to be biocompatible, biodegradable where the degradation rate should be correlated with the regeneration time of the new tissue, and should have interconnected pores of appropriate size allowing cell attachment, migration, proliferation and vascularization to supply the necessary nutrients to the newly formed tissue [391]. Most of the hydrogels used for wound dressing are usually loaded with drug to increase the rate of healing of the skin. Shamloo et al. [214] prepared a PVA/chitosan/gelatin hydrogel embedded with PCL microspheres by a double-emulsion-solvent-evaporation method destined for accelerated wound healing by sustained release of bFGF. Electron beam irradiation (EBI) was used to prepare hybrid keratin based hydrogels in the presence of two synthetic polymers, namely PVA and polyethylenimine (PEI) that influence the gelation rate of keratin [392]. In this work, keratin was extracted from two natural sources, human hair and wool, using sulfitolysis reaction with sodium disulfite in order to cleave the cystine disulfide bonds and to form cysteine thiol. Because during EBI, the aqueous solution of obtained reduced keratin was not converted to gel, PVA was added to the solution inducing gel formation at an EBI dose of approximately 90 kGy. Besides, by the addition of a secondary synthetic polymer, namelly PEI, in the aqueous blend containing S-sulfo keratin and PVA, the gelation occurs at a much lower irradiation dose, up to 10 kGy. The gel forming may be assigned not only to the physical interaction between S-sulfo keratin chains, oxygen groups in PVA aqueous solution and the amine groups of PEI, but also to the keratin covalent crosslinking by EBI. Later, these hydrogels have been tested for wound healing [393] and was suggested that the treatment with these keratin-based hydrogels enhanced the production of new collagen and fibroblast proliferation during granulation tissue formation and the remodeling phase of wound healing. Synthetic polymers as PEGDA [394], PVA [395], poly(N-hydroxyethyl acrylamide) (PHEA) [396], PAAm [397] etc., were used in combination with silk sericin to obtain hybrid hydrogels that could be used especially for wound dressings and dermal reconstructions.

#### 7.3.5. Drug and Molecule Delivery

Due to its biocompatibility and well-established safety profiles, hydrogels fabricated using collagen have been employed as delivery vehicles for therapeutic genes, which can direct and/or enhance the function of the transplanted or endogenous cells [398].

Krebs et al. [399] showed the collagen chains ability to locally load siRNA (small interfering RNA) and then release it in a controlled manner in order to extend the effect directly at the specific target. Peng et al. [400] tested collagen based hybrid hydrogels for the Id1-siRNA targeting and its delivery and sustained release for the treatment of gastric cancer. To improve the siRNA delivery by stimulating the target of Id1-siRNA into the defect cells and prolonging the silencing effect, cationic PEI was further engaged for scaffold modification.

Gelatin has been used for nanogels preparation for different biomedical applications [401]. In this respect, activated PEG methyl ether was grafted onto gelatin backbone to obtain the gelatin-PEG copolymer and then self-assembled to form nanogels (Figure 8) that were encapsulated with poor water soluble curcumin (CUR) for cancer treatment [402]. Nanogels improved the solubility of CUR in the aqueous environment, protected it from the hydrolytic degradation and also increased the amount of delivered CUR in a control release manner, thus enhancing the therapeutic efficacy of CUR.

Gelatins have been used in designing of different formulations suitable as vectors for gene delivery [403,404]. Rumschöttel et al. [405] were incorporated core-shell like spherical DNA/PEI polyplexes in gelatin hydrogels. After a μ-DSC study, it has concluded that by adding a third component, e.g., gelatin, the complex stability between the polycation PEI and the negatively charged DNA is influenced by its interaction with the polycation shell. Thus, the effect of the gelatin can be explained by the weakening of the electrostatic interactions in DNA/PEI polyplexes, due to the gelatin attachment to the PEI shell or partially takes off the PEI from the shell. Conversely, in the DNA/maltose-modified PEI polyplexes, a stronger interaction via H-bonding between maltose units and gelatin was confirmed.

A pH sensitive and partly-biodegradable sIPN hydrogel based on methacrylic acid, BSA, and PEG, with potential application as drug delivery systems, was prepared by UV initiated free radical polymerization [406]. Additionally, BSA was copolymerized with PEG to obtain high-water content (>96%) hydrogels [407] that be useful for the preparation of controlled release devices in the field of wound dressing.

In the field of controlled and sustained drug delivery systems, SF based hydrogels were tested using as model drugs curcumin [408] and hydrophobic/hydrophilic drugs, namely aspirin and indomethacin [409], where the synthetic part was represented by PVA and PLA-PEG-PLA copolymer, respectively.

WPI based hydrogels can be used as bioresponsive carriers for controlled release of biomolecules and drugs [410], as they exhibit good pH-sensitivity and protects the entrapped drug from degradation by its grafting onto synthetic polymers resulting co-polymers. An example was reported by Aderibigbe and Ndwabu [411] who prepared by simultaneous redox cross-linked polymerization an WPI-*g*-carbopol polyacrylamide based hydrogel loaded with pamidronate, (nitrogen containing bisphosphonate), to treat the skeletal disorders in children, osteoporosis and bone cancer. The hybrid hydrogels showed a pH dependent swelling behavior, being more swollen at high pH values.

## 8. Nucleic Acids Based Hybrid Hydrogels for Biomedical Applications

Deoxyribonucleic acid (DNA) and ribonucleic acid (RNA) are polynucleotides with distinct biological function. The specific bonding of DNA base pairs (for example guanine–cytosine, adenine–thymine for DNA and adenine–uracil in the case of RNA) provide the chemical foundation for genetics, being a powerful molecular recognition system [412,413]. The level of versatility and structural programmability of nucleic acids ranks them on a higher level in the field of biomedical applications compared to other natural or synthetic polymers. Since 2001 (according to SCOPUS database) there has been an intensification of research in the field of nucleic acids-based hydrogel development, especially due to the great progress in DNA synthesis and consequently the accessibility of larger amounts of DNA. Due to the outstanding properties of nucleic acids as hydrophilicity, biocompatibility, stimuli responsiveness, versatility and structural programmability, controlled biodegradability, which can be exploited in the development of new materials for numerous biomedical applications, this may be the subject of a separate review. In the recent published scientific literature there are already comprehensive reviews covering all aspects of obtaining and using hydrogels based on or containing nucleic acids in biomedical applications [414,415,416]. Hence, in the current review is presented just a short summary regarding the development of nucleic acids-containing hydrogels.

### 8.1. Nucleic Acids Ability to Form Hydrogels

DNA is an outstanding component for the obtaining of supramolecular hydrogels, especially because it is an amphiphilic bio-polylectrolyte capable to absorb large quantities of water and may induce in the incorporating material controllable properties, programmability and biocompatibility [415]. At this moment there are three approaches for using nucleic acids in the construction of hydrogels, namely: (1) as a building unit for hydrogel-based materials—representing the so called “all-DNA” hydrogels [417,418,419]; (2) as biocompatible cross-linkers combining nucleic acids with synthetic polymers, resulting hybrid hydrogels [420]; and (3) short DNA sequences (e.g., aptamers) that act as functional grafts to obtain tailorable hydrogels with unique bio-specificity [414,421,422].

Even though DNA-based hydrogels have numerous distinctive properties, their use in the biomedical field beyond conceptual demonstration is limited by a number of shortcomings, namely: small number of functional groups available (needed in further chemical modifications), highly negative net charge, and high costs of synthesis [414]. To overcome these drawbacks, the combination of nucleic acids with hydrophilic polymer scaffolds represents a promising approach. Polyamines show the ability to bind DNA by promoting its condensation, a process of paramount importance in medical applications, mainly in immunotherapy. Among the synthetic polyamines, poly(ethylene imine) (PEI) is one of the most prominent examples of cationic polymers capable to induce condensation of DNA [423]. PEI has a high charge density and by electrostatic interactions it can strongly bind to DNA leading to its compaction [415,424].

### 8.2. Biomedical Applications of Nucleic Acids-Containing Hybrid Hydrogels

#### 8.2.1. Drug Delivery

Spherical DNA-based nanogels were synthesized by Costa et al. [425] using PEI as pDNA condensation agent. These nanogels showed very low size distribution, high loading capacity and interesting release kinetics of simultaneous delivery of pDNA and drugs. In a recent study, DNA was grafted with PCL to obtain hybrid nanogels that were then loaded with Cas9/single guide RNA complex. The resulted DNA based hybrid hydrogel can be used as carrier for gene editing tool delivery that provides excellent physiological stability against nuclease digestion [426].

#### 8.2.2. Immunotherapy

Mimi et al. [427] have fabricated a gelatin PEI core shell nanogel via a two-stage synthesis. The resultant nanogels are highly uniform spherical particles and have a well-defined core shell nanostructure with a biodegradable gelatin core and a hairy and extended PEI shell. The resultant nanogels were able to completely condense siRNA, forming stable complexes that were capable of protecting the siRNA from enzymatic degradation. The gelatin PEI nanogels were four times less toxic than the native PEI, and were able to effectively deliver the siRNA into HeLa cells. Increasing the N/P ratio significantly improved the intracellular uptake efficiency of the siRNA. Li et al. [428] has proposed a flexible strategy to design well-defined reducible cationic nanogels (PGED-NGs) based on ethylenediamine (ED)-functionalized low-molecular-weight poly(glycidyl methacrylate) (PGMA) with friendly crosslinking reagents (α-lipoic acid). PGED-NGs could effectively complex pDNA and siRNA. Compared with pristine PGED, PGED-NGs exhibited much better performance of pDNA transfection. PGED-NGs also could efficiently transport metastasis-associated lung adenocarcinoma transcript 1 siRNA into hepatoma cells and significantly suppressed cancer cell proliferation and migration. After the photoisomerization of the Azo moiety on the DNA cross-linker, Kang et al. [429] succeded to develop photoresponsive hydrogels based on comb-shaped DNA-polyacrylamide conjugates that can be used for the DOX delivery in cancer therapy. More precisely, these hybrid hydrogels were capable to carry and then to release a large amount of active drug molecules (DOX) in a controllable manner due to their ability to transform in sol upon UV light irradiation, and subsequent inducing a very high rate of cancer cell death. siRNA based nanoparticle complexes have been embedded in PEG-PLA-dimethacrylate hydrogels in order to obtain biodegradable systems that exhibits high controlled tissue-specific localization and sustained gene delivery due to the hydrolytic degradation of ester bonds within the PLA crosslinks [430]. Negatively-charged siRNA was complexed to nanoparticles by its self-assembly with a poly(dimethylaminoethyl methacrylate)-*b*-poly-(dimethylaminoethyl methacrylate-*co*-propylacrylic acid-*co*-butyl methacrylate) diblock polymer.

#### 8.2.3. Biosensing Applications

The analyte-triggered opening of the DNA-crosslinked structures and consequent dissociation of the gel networks can initiate the release of encapsulated signal substances to generate an output signal. Using different responsive DNA hydrogels in which enzymes, DNAzymes, or catalytic nanomaterials are entrapped, researchers have developed various colorimetric visual sensors and readout devices based on the sol-gel transition strategy for different kinds of biotargets, such as metal ions and glucose. Liu and co-workers [431] demonstrated that covalently crosslinked polyacrylamide hydrogels can be used as a platform to attach Hg^2+^-responsive DNA structures for the ultrasensitive detection and removal of Hg^2+^ [432]. Shape-memory DNA hydrogels that can reversibly respond to external stimuli have also been developed. Hu et al. designed bilayer hybrid hydrogels whose stiffness can be controlled by different triggers such us thermal and pH (i-motif) stimuli [433]. One of the two layers was consisted of a non-responsive hydrogel based on acrydite-DNA crosslinked polyacrylamide, and the second layer included a thermosensitive acrydite-DNA crosslinked PNIPAM based hydrogel. The two-layer hybrid hydrogels that exhibits controlled stimuli-induced shape transitions are promising for the use in the biomedical field as intelligent actuators. Similar studies, of shape memory hydrogels that include DNA and polyacrylamide were found in the scientific literature [434,435]. In Figure 9 is illustrated a photoresponsive hybrid hydrogel that reveals light-induced switchable stiffness functions. The hybrid hydrogel is based on pDNA crosslinked with polyacrylamide and stabilized by trans-azobenzene intercalator units.

## 9. Hybrid Hydrogels Containing Lignin for Biomedical Applications

Lignin hydrogels are considered to have a great potential for valorization of this abundant polyaromatic bio-polymer abundant in plants and which results as by-product in pulp and paper industry. The three phenolic sub-structures of the lignin structure, namely syringyl (S), guaiacyl (G), and p-hydroxyphenyl (H) units and also lignin derivatives (resulted by epoxidation, amination, hydroxyalkylation, nitration, halogenation, sulfomethylation, etc.) contain many different functional groups (hydroxyls, carboxyls, carbonyls and methoxyls) as active sites, in preparation of functional hydrogels both by physical crosslinking with hydrophilic polymers by H-bonding and also by, polymerization, copolymerization, ATRP and reversible addition-fragmentation transfer (RAFT) polymerization, chemical crosslinking, crosslinking grafted lignin and monomers etc. [436,437,438].

Lignin-based hydrogels had a rougher surface morphology and a porous structure related to an increasing lignin concentration. The water uptake and the water retention of lignin-based hydrogels depend on structure of the hydrogels, including the pores sizes and the surface morphology. Physical hybrid hydrogels showed self-healing capability due to the dynamic hydrogen bonding between lignin and hydrophilic polymers.

Increasing the content of lignin in hydrogels significantly improved the mechanical properties of the hydrogels [439] and they show excellent strength properties. The biodegradation of the lignin-based hydrogel depends on the crosslink density and the phenolics content in the hydrogel. The former is correlated with small pores, stronger cross-linked gels which had less accessibility for lignolytic fungi and actinomycetes whch induces high resistance against microbial attack than slightly cross-linked hydrogels. The phenolic substructures in hydrogels directly attacked the expressed enzyme systems of lignolytic fungi [440] so they protect plant from fungi invasion. Hydrogels containing lignin and its derivatives with acrylic acid, methacrylic and NIPAAm monomers exhibit external stimuli response as pH response, thermoresponsiveness and mechanical response. The self-healing property of lignin-based supramolecular lignin-based hydrogel with α-cyclodextrin and a hyper branch architecture was obtained that possessed a mechanical response feature. When the oscillation stain increased, the solid-like hydrogel translated into liquid-like, and the liquid-like hydrogel recovered to be solid-like via self-assembly when the oscillate stain decreased [441].

The biocompatibility, biodegradability, low toxicity and eco-friendliness of lignin-based hydrogels are main features which have determined their wide application as biomaterials in many fields as for controlled release of the functional materials including enzyme immobilization and drug delivery.

### 9.1. Bioactive Compounds Delivery

Lignin-based hydrogels could be used as biological carriers for human hepatocytes culture, because a large number of hepatocytes adhered to the pores of the lignin-based hydrogel, and a higher cell proliferation rate and metabolic activity was reported [442]. From lignin grafted polymers and hydrophilic polyurethane/acrylic monomer, a double-network hydrogel with tough and pH sensitive properties [443] was prepared and can be processed by fiber spinning, casting and 3D printing, this one being biocompatible with primary human dermal fibroblasts [444]. Lipase immobilized on cellulose/lignin hydrogel beads exhibited higher stability and activity [445].

A hydrophilic supramolecular complex of hemicellulose/oak lignin hydrogel with pectin embedded in the 3D structure was prepared and tested to deliver β-glucuronidase and estrogens [446].

A smart hydrogel of carboxylated lignin nanoparticles/PEG-poly(histidine) block copolymer/a cell-penetrating peptide loaded with a poorly water-soluble cytotoxic agent was evaluated for chemotherapeutic potential [447]. The reactivity of lignin with epichlorohydrine as curing agent was exploited to obtain superabsorbent hydrogels with cellulose and its derivatives, PVA, xanthan, which controlled release active substances as phenols and vanillin, active aroma ingredient, [28,448,449].

Acid activated lignosulphonate/poly(vinyl pyrrolidone) hydrogel is a good carrier for controlled release of amoxicillin [450].

### 9.2. Applications as Antimicrobial, Antioxidant, Antifungal materials

Lignin itself, as a complex natural polymer with phenolic groups, possess non-toxicity, antimicrobial, antioxidant and antifungal properties, which may enhance the potential application in food science and health care of its hydrogels [451,452]. Antibacterial and antioxidant agents like CS and Alg had been added into lignin-based hydrogels, which had great potential to be applied in biosensors and tissue engineering [220].

Modification of the lignin with triazole moiety enhances the antibiofilm and antimicrobial activities. A new hydrogel was prepared as an ointment for anti-infection, which had abilities to prevent infection of burn wound and could be used as an anti-inflammatory dressing and aid-healing material [453]. Lignin/CS hydrogels show a good potential applicability in wound healing, since these hydrogels present a good cell attachment and proliferation of NIH 3T3 mouse fibroblast [454]. Highly antibacterial lignin-based hydrogels for drug delivery have been recently reported against *Staphylococcus aureus* and *Proteus mirabilis*, behaviour demonstrated for hydrogels prepared with lignin, cellulose, hyaluronan, Gantrez S-97 (poly(methyl vinyl ether-*co*-maleic acid), poly(ethylene glycol) or glycerol and others [455,456,457].

## 10. Conclusions and Future Trends

Because of the large availability of materials used in organic hybrid polymeric hydrogels (both synthetic and natural polymers), the number of possible combinations is enormous, which explained the large number of publications only in the last decades (more than 2000 annually). The biomedical applications of the hydrogels started back multiple decades and their study is continuously growing.

In this review, some selective research studies have been summarized especially in the last two decades, for the preparation of natural polymers-containing hybrid hydrogels and their potential application in a wide range of medical applications. It was described both advantages and disadvantages of each hydrogel applied in different medical application. Desired hybrid hydrogels may be developed for targeted applications by making changes in composition, use of specific biomolecules, antimicrobial agents, use of suitable cells, and selecting suitable synthesis routes and processing techniques. The successful use of a polymeric hybrid hydrogel consists in creating a three-dimensional micro-/nano environment that represents a synthetic ECM for the cells, which should provide biodegradability, biocompatibility, pore interconnectivity to assure the penetration and absorption of nutrient, modulation of proliferation for successful reconstruction of organs, cell-adhesion and regeneration certain tissue. In the most recent researches, injectable hydrogels and 3D-bioprinted hybrid hydrogels allow successful their interaction with the cells of damaged tissues. The hybrid nano hydrogel materials are able to convert external stimuli signals to heat, highly oxidative species etc., which are helpful for combinatorial therapies and theranostics. By a simple hybridization of the components of the hybrid hydrogels smart multiresponsive materials can be obtained by synergistic combination of the best properties of both components, useful toward applications in nanomedicine which exhibit an excellent targetability, minimal side effects in treatments and diagnostic. The industrial application of the new hybrid hydro/nanogels materials is in its first steps and it need more relevant clinical data concerning their safety and efficacy in vivo. In Table 6 are listed few examples of hybrid nanogels/hydrogels evaluated in preclinical and clinical studies.

CHP nanogels have been used as an intranasal vaccine-delivery system [459,462,464] their clinical trials showed promising results for therapy-refractory esophageal cancer patients and HER2 expressing cancer patients. The CHPHER2 complex vaccine was safe, well tolerated and showed specific antibody responses (Phase I and II of trial). From the products already on the market that use a combination, one can mention the HYAFFTM esterified hyaluronic acid [465], produced by FIDIA Ltd.: LaserskinAutograft^®^ (made of a HA-membrane with keratinocytes), and Hyalo-graft 3D^®^ (made with HA, but with fibroblasts added). More preclinical studies are needed to provide convincing resources for supporting advanced clinical applications.

According to the Food and Agriculture Organization of the United Nations and World Health Organization, probiotics are “live microorganisms which, when administered in adequate amounts, confer a health benefit on the host”. For probiotic delivery systems, biomaterials such as proteins (gelatin, casein or whey proteins), polysaccharides, as well as synthetic polymers, such as poly(d,l-lactic-*co*-glycolic acid), polyvinyl alcohol or Eudragit (poly(methacrylic acid-*co*-ethyl acrylate) 1:1) could be used [466].

The complex (synthetic and natural) polymeric hybrid hydrogels with functional domains or nano/microstructures that provide both improved mechanical and physical properties, tunable release kinetics for targeted drug therapy, mediated cell response, stimuli-responsive material behavior are under continuously development. Their use in clinical and research applications in biomedical practice, as drug discovery, drug/gene delivery, regenerative medicine, etc., is very promising. New and/or improve existing multi length-scale methodologies for predicting the properties of the materials based on those of the individual components, as a guide for experimental development is necessary. A better understanding of the interactions between components of the hybrid hydrogels will lead to a more efficient design and control of their mechanical performance, long-term stability, the hydrophilic/hydrophobic nature of the material for improved drug loading capacity and controlled release, or the precise control of shape variations induced by external stimuli, to name a few. They are used to impart to homopolysaccharides-based hydrogels stimuli responsiveness (e.g., pH, temperature), to improve their mechanical properties, etc. Is noticed a tendency of the research towards the variation of the methods of modifying the polysaccharides properties (by obtaining new copolymers, functionalization through various chemical reactions, using innovative techniques such as 3D printing or electrospinning) to meet the specific requirements of biomedical applications. Using composite hydrogels containing various inorganic nanoparticle is another direction with multiple possibilities to obtain materials with tunable properties and targeting applications directed to personalized medicine. This are/will be subject to other many reviews.

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
