# Peer review of "New Developments in Medical Applications of Hybrid Hydrogels Containing Natural Polymers"

_molecules, 2020, doi:10.3390/molecules25071539_

Round 1
Reviewer 1 Report
The review paper summarizes the developments on the biomedical applications of hybrid polymeric hydrogels containing natural polymers and synthetic polymers. The content is systematic and quite in detail. However, there is an obvious drawback of the manuscript, that is, the lack of attractable figures. There are only 3 pictures in the paper whereas they are all with poor quality. Especially for a review paper, figures would not only favorably descript the designation of a cited work or the advantages of the materials being discussed in by authors, but are the key objects for gaining high impact of the manuscript from the readership. Therefore, I suggest the authors to revise the manuscript with suitable supplementary of figures. Besides, there are many typos in the text. Just for an example, in Scheme 1, “shyntetic” polymer, should it be synthetic polymer? In addition, also in this scheme, it classifies “particle size” as microgel or nanogel, which is very confusing. What the “particle size” representing? It should not describe the physical appearance as this has been mentioned in the same scheme. Even in the text, it is not clearly mentioned. The authors are thus asked to more carefully read and check the whole manuscript during the revision.
Author Response
Thanks to the reviewer for the valuable comments. The answers are attached
The authors

Reviewer 2 Report
The authors have written a very comprehensive review that runs to some 60 pages that summarises key findings in most of the different classes of hydrogel materials. Generally, the review is well-organised. I recommend acceptance of the article after some changes are made:
- Title can be improved. I suggest 'New developments in hybrid hydrogels containing natural polymers for medical applications'.
- The abstract needs to be re-written. The first sentence basically repeats what the title mentions, and the second sentence starting from line 15 is 8 lines long, which is clearly unacceptable and really difficult to read. Please consider rephrasing.
- Line 22: what does 'a.s.o' mean?
- Instead of section 1 being 'definitions and classification', I will much prefer to see an actual 'Introduction' which introduces the field to readers and lays the context for the rest of the review.
- Despite the long review, there are very few figures which help understanding and make it easier for the reader. Consider adding some diagrams as well at crucial junctures. For example, Section 2 will benefit from diagrams explaining the different synthetic routes. There are many more areas as well.
- Line 184: 'Synthesising' should be 'synthetic'
- Pg 7: 'processing methods' and '3D bioprinting' are not considered synthetic routes, as there are no actual covalent bonds being formed. Please remove from this section, or distinguish it from the other routes, as it can be misleading.
- Line 351: 'shyntesis' is a spelling mistake
- A new interesting application of chitosan-based hydrogels in treating oral ulcers has been reported 'https://pubs.rsc.org/en/content/articlelanding/2020/bm/c9bm01754b/unauth#!divAbstract' I suggest citing and discussing this article for completeness.
- I was slightly disappointed when I found out that there was no section on 'Polynucleotides and others'. Given the large number of hydrogels based on DNA/ RNA molecules, as well as lignin, please add in a section discussing these hydrogels from these molecules as well. Lignin will be especially important in light of today's sustainability trends, and will enable the review to achieve higher impact.
Author Response

(The authors gave the same response as above.)

Round 2
Reviewer 1 Report
The manuscript has been properly revised and therefore is recommended for publication.